# Disulfiram Ophthalmic Solution Inhibited Macrophage Infiltration by Suppressing Macrophage Pseudopodia Formation in a Rat Corneal Alkali Burn Model

**DOI:** 10.3390/ijms24010735

**Published:** 2023-01-01

**Authors:** Toyo Ikebukuro, Takeshi Arima, Momoko Kasamatsu, Yuji Nakano, Yutaro Tobita, Masaaki Uchiyama, Yuya Terashima, Etsuko Toda, Akira Shimizu, Hiroshi Takahashi

**Affiliations:** 1Department of Ophthalmology, Nippon Medical School, Tokyo 113-8603, Japan; 2Department of Analytic Human Pathology, Nippon Medical School, Tokyo 113-8603, Japan; 3Division of Molecular Regulation of Inflammatory and Immune Diseases, Research Institute for Biomedical Sciences, Tokyo University of Science, Chiba 278-0022, Japan

**Keywords:** disulfiram, FROUNT, corneal inflammation, corneal scarring, anti-neovascular, macrophage infiltration, alkali burn, LV-SEM

## Abstract

FROUNT is an intracellular protein that promotes pseudopodia formation by binding to the chemokine receptors CCR2 and CCR5 on macrophages. Recently, disulfiram (DSF), a drug treatment for alcoholism, was found to have FROUNT inhibitory activity. In this study, we investigated the effect of DSF eye drops in a rat corneal alkali burn model. After alkali burn, 0.5% DSF eye drops (DSF group) and vehicle eye drops (Vehicle group) were administered twice daily. Immunohistochemical observations and real-time reverse transcription-polymerase chain reaction (RT-PCR) analyses were performed at 6 h and 1, 4, and 7 days after alkali burn. Results showed a significant decrease in macrophage accumulation in the cornea in the DSF group, but no difference in neutrophils. RT-PCR showed decreased expression of macrophage-associated cytokines in the DSF group. Corneal scarring and neovascularization were also suppressed in the DSF group. Low-vacuum scanning electron microscopy imaging showed that macrophage length was significantly shorter in the DSF group, reflecting the reduced extension of pseudopodia. These results suggest that DSF inhibited macrophage infiltration by suppressing macrophage pseudopodia formation.

## 1. Introduction

C-C chemokine receptor types 2 (CCR2) and C-C chemokine receptor types 5 (CCR5) and their respective ligands play important roles in the recruitment of monocytes and macrophages to inflammation sites [1,2,3,4,5,6,7,8,9,10,11]. A protein called FROUNT (NUP85, Nucleoporin85) was shown to promote the phosphatidylinositol-3-OH kinase (PI3K) cascade and cell migration by binding directly to the intracellular C-terminal region of CCR2 and CCR5 [12,13,14,15,16]. Disulfiram (DSF) was identified in a drug screen as a FROUNT inhibitor; DSF binds directly to a specific site on FROUNT and inhibits FROUNT-CCR2 and -CCR5 interactions [17]. DSF has been used as a treatment for alcoholism since the 1940s. In recent years, it has attracted attention for its many reported inhibitory effects on cancer and inflammation [18,19,20]. Our group has found inhibitory effects on cancer-associated macrophages and macrophages in glomerulonephritis in relation to FROUNT inhibition [17,21]. In the ophthalmology field, DSF eye drops have been studied in a rat model of uveitis but there are no reports regarding corneal scarring and angiogenesis [22]. In this study, we investigated the effects of DSF eye drops on macrophage infiltration, corneal scarring, and angiogenesis in a rat corneal alkali burn model [23,24,25]. The alkali burn model is a model for evaluating corneal inflammation and wound healing by inflicting artificial alkali burns on the cornea. After alkali burn, 0.5% DSF eye drops (DSF group) and vehicle eye drops (Vehicle group) were administered twice daily. Immunohistochemical observations and real-time reverse transcription-polymerase chain reaction (RT-PCR) analyses were performed at four endpoints: 6 h and 1, 4, and 7 days after alkali burn. We also measured the length of macrophages using a low-vacuum scanning electron microscope (LV-SEM) to evaluate the formation of macrophage pseudopodia in the corneal tissue [26,27,28,29].

## 2. Results

### 2.1. Corneal Scarring after Alkali Burn

Macroscopic images of the cornea at the time of alkali burn and 7 days after twice-daily administration of Vehicle or DSF eye drops after alkali burn are shown (Figure 1a–e). Light reflection could be clearly observed in the cornea before alkali burn. The iris was contracting and the iris vessels were seen to extend radially (Figure 1a). Alkali burns were created by placing filter paper soaked in NaOH at the center of the cornea (Figure 1b). Immediately after alkali burn, a part of the iris blood vessels was obscured, and corneal opacity was seen in the center of the cornea (Figure 1c). At 7 days after alkali burn, the cornea of the Vehicle group lost transparency and the reflection of light from the microscope was unclear. The iris was adherent and unable to contract. Peripheral vascularity was increased compared to that before alkali burn, suggesting the possibility of neovascularization in the periphery of the cornea and iris (Figure 1d). In the DSF group, transparency was preserved, although there was mild opacity. The iris was the same as before alkali burn and retained its shape and was contracted (Figure 1e). Hematoxylin and eosin (HE) staining was used to evaluate the corneal epithelium of the central cornea. At 6 h after alkali burn, both groups showed defects of corneal epithelium, but the DSF group had fewer defects than the Vehicle group (Figure 1f,g). At 1 day, both the Vehicle and DSF groups had complete epithelial coverage (Figure 1h,i).

### 2.2. Evaluation of Macrophage Infiltration with Disulfiram Eye Drops

To evaluate macrophage infiltration in the cornea, CD68 antibody (ED-1) immunostaining was performed. ED-1-positive cells were observed to infiltrate the peripheral cornea 6 h after alkali burn and the central cornea 1 day later. ED-1-positive cells were counted at four endpoints (6 h and 1, 4, and 7 days) for each of the peripheral and central corneas, and the Vehicle and DSF groups were compared (Figure 2a–p). The results showed that macrophage infiltration was suppressed in the DSF group in both the peripheral and central areas. In the peripheral cornea, significantly fewer ED-1-positive cells were observed in the DSF group after 1 day (Figure 2q). In the central cornea, ED-1-positive cells are absent at 6 h but infiltrated after 1 day. After 4 days, there were significantly fewer ED-1-positive cells in the DSF group (Figure 2r).

### 2.3. Evaluation of Neutrophil Infiltration with Disulfiram Eye Drops

To evaluate neutrophil infiltration to the cornea, naphthol AS-D chloroacetate esterase (EST) staining was performed. EST-positive cells were counted at four endpoints (6 h and 1, 4, and 7 days) for each of the peripheral and central corneas, and the Vehicle and DSF groups were compared (Figure 3a–p). EST-positive cells infiltrated the peripheral cornea starting at 6 h and peaked at 1 day; they rapidly disappeared after 4 days (Figure 3q). In the central cornea, EST-positive cells were observed after 1 day, but the number of cells that reached the cornea was already quite low and almost none were observed after 4 days (Figure 3r). Neutrophils infiltrated early and decreased early in both groups. Cell counts showed no significant difference in the number of EST-positive cells between the two groups (Figure 3q,r).

### 2.4. Inhibition of Macrophage-Associated Cytokines by Disulfiram Eye Drops

Fluorescence immunohistochemical staining was used to examine the presence of FROUNT and phosphorylated AKT, a molecule activated in the PI3K pathway (in which FROUNT was reported to be involved), in ED-1-positive cells infiltrated in a corneal alkali burn model (Figure 4a,b). We confirmed the presence of macrophage-associated cytokines, TNF-α, TGF-β1, and IL-1β, on ED-1-positive cells infiltrated in the corneal tissue (Figure 4c–e). RT-PCR was performed on these cytokines using corneal tissue from the DSF and Vehicle groups isolated 4 days after alkali burn (Figure 4f–h). The results showed that the expression of these macrophage-associated cytokines was significantly decreased in the DSF group.

### 2.5. Inhibition of Corneal Scarring by Disulfiram Eye Drops

To evaluate scarring of the central cornea 7 days after alkali burn, immunohistochemical analysis of collagen type III was performed. Collagen type III was found in the corneal stroma 7 days after burn in both the Vehicle and DSF groups (Figure 5a,b). In the Vehicle group, there were many strongly stained areas near the superficial layer of the corneal stroma and many gaps between collagen fibers. In contrast, the DSF group showed less strongly stained stroma and fewer gaps between collagen fibers. The percentage of collagen type III area in the corneal stroma was significantly decreased in the DSF group compared to the Vehicle group (Figure 5c). RT-PCR at 4 days after alkali burn showed that TGF-β1, a cytokine associated with fibrosis, was significantly downregulated in the DSF group (Figure 4g).

### 2.6. Inhibition of Angiogenesis by Disulfiram Eye Drops

To evaluate neovascularization in the peripheral cornea, immunostaining for nestin, which indicates proliferating vascular endothelial cells, was performed on the corneas of rats 4 days after alkali burn. In the Vehicle group, numerous neovascular vessels were observed in the peripheral cornea (Figure 6a). On the other hand, neovascular vessels were also observed in the DSF group, but less so than in the Vehicle group (Figure 6b). Positive cell counts in both groups revealed that the DSF group had significantly fewer nestin-positive cells than the Vehicle group (Figure 6c). Vascular endothelial growth factor A (VEGF-A) was also examined using RT-PCR. The results showed that VEGF-A expression was significantly decreased in the DSF group compared to the Vehicle group (Figure 6d). The expression of TNF-α and IL-1β, which increases the expression of VEGF, was also significantly decreased in the DSF group (Figure 4f,h).

### 2.7. Evaluation of Macrophage Pseudopodia Formation Using LV-SEM

DSF inhibits the pseudopodia formation of macrophages by blocking FROUNT and suppressing their infiltration and migration [17]. Therefore, evaluation of pseudopodia formation is important to assess the involvement of FROUNT in the inhibition mechanism of macrophage infiltration. Since macrophages become longer when they extend their pseudopodia, we indirectly evaluated macrophage pseudopodia formation by measuring the length of macrophages infiltrating the peripheral cornea. LV-SEM was used to ensure detailed morphological observation at high magnification and measurement accuracy. ED-1 immunostaining was performed on rat corneas 1 day after corneal alkali burn to detect macrophages, and heavy metal staining was applied to enhance the signal and facilitate observation of ED-1-positive macrophages. Corneal infiltrating macrophages in the Vehicle group exhibited a spindle-like, elongated shape along the fibers of the cornea. In contrast, macrophages in the DSF group had a less elongated and more rounded shape compared to the Vehicle group (Figure 7a–d). In both the Vehicle and DSF groups, the length of 392 ED-1-positive cells were measured from 11 eyes each. The results showed that the length of ED-1-positive cells was significantly shorter in the DSF group than in the Vehicle group (Figure 7e).

## 3. Discussion

Corneal damage due to a chemical burn causes corneal epithelial loss and inflammatory cell infiltration in the acute phase, while corneal scarring and neovascularization are long-term problems. In ophthalmology, it is one of the most serious diseases that worsen visual prognosis. [23,24,25]. Early anti-inflammatory treatment is important, and steroid eye drops are often used in clinical practice. However, some patients have difficulty with long-term use of steroid eye drops due to their inherent side effects, such as increased intraocular pressure [30]. Therefore, anti-inflammatory eye drops that can be used as a substitute for steroids or in combination with steroids are desired. Our group showed that DSF administered in a murine tumor model targeting FROUNT inhibited macrophage infiltration in the tumor tissue [17]. In this study, we examined the effect of DSF on corneal alkali burn, focusing on the inhibition of macrophage infiltration.

DSF inhibits the interaction between CCR2/CCR5 and FROUNT, thereby suppressing macrophage pseudopodia formation and tissue infiltration. Although there are reports that neutrophils express CCR2 and CCR5 under specific conditions, it is generally considered that these receptors are expressed on macrophages, monocytes, lymphocytes, and dendritic cells, but not neutrophils [1,31]. In this study, as expected, cell counts showed a significant reduction in infiltrating macrophages in the DSF group, while no difference was observed in neutrophils between the two groups. This experimental result of macrophage-specific inhibition supports a FROUNT-mediated mechanism. On the other hand, the lack of effects on neutrophils in the present study suggests that FROUNT may not be effective against hyperacute inflammation, in which neutrophils are mainly involved.

Since DSF only inhibits the interaction between FROUNT and CCR2/CCR5 and does not decrease FROUNT, it is not possible to measure the effect of DSF by quantifying FROUNT. In this model, infiltrating macrophages in the cornea after alkali burn express FROUNT and phosphorylated Akt, which is reported to be activated through FROUNT, suggesting the involvement of FROUNT-mediated mechanisms in macrophage infiltration in the injured cornea [17]. In addition, macrophage-associated cytokines, TNF-α, TGF-β1, and IL-1β, were confirmed to be present on infiltrating macrophages, and expression levels in the two groups were examined using RT-PCR, with significantly decreased expression in the DSF group. When combined with the results of macrophage cell counts, it is likely that these cytokines were reduced due to the suppression of macrophage infiltration. 

Collagen type III is weakly expressed in the normal corneal stroma, but its expression increases in the injured cornea. As a result, corneal transparency is reduced [32]. In the present study, the area positive for collagen type III showed a significant decrease in the DSF group compared to the Vehicle group, and RT-PCR showed decreased expression of TGF-β1, which promotes fibrosis [33,34,35,36,37]. These results, together with the macrophage cell count results, suggest that a decrease in TGF-β1 following a decrease in infiltrating macrophages may have suppressed fibrosis. Macrophages include M1 macrophages, which are involved in inflammation, and M2 macrophages, which regulate immunity and suppress inflammation [38,39]. In this study, we enumerated pan-macrophages (including both M1 and M2 macrophages); thus, the effect of DSF eye drops on M2 macrophages is unknown. Previous DSF reports on glomeruli have shown that M2 macrophages were reduced, but fibrosis was suppressed [21]. M2 macrophages should also be evaluated in a corneal alkali burn model in the future.

Here, DSF eye drops decreased the number of vascular endothelial cells positive for nestin. Nestin is a marker of neovascularization that is observed in proliferating vascular endothelial cells [40]. Neovascularization of the cornea causes loss of corneal transparency and reduced visual acuity. VEGF-A is an essential factor for neovascularization, and suppressing VEGF-A is very important in preventing corneal neovascularization [41,42]. RT-PCR analysis showed that VEGF-A decreased in the DSF group. Previous studies have reported that the macrophage-associated cytokines TNF-α and IL-1β both enhance VEGF-A expression [43,44]. RT-PCR analysis demonstrated that these cytokines were also downregulated in the DSF group, and may have indirectly influenced the decreased expression of VEGF-A. These results, together with the macrophage cell count results, suggest that a decrease in VEGF-A as a result of a decrease in infiltrating macrophages may have suppressed angiogenesis. In this experiment with rat corneas, neovascularization was observed as early as 4 days, but neovascularization that occurs in human corneal trauma is not usually observed within a few days. Therefore, although DSF is expected to suppress neovascularization, the present results cannot be directly extrapolated to complications in human occurring in the later stages, and further validation is needed.

LV-SEM is commonly used in the renal glomerular region and has rarely been reported in the field of ophthalmology. Our group previously employed LV-SEM to evaluate corneal wound healing in a rat corneal alkali burn model [28]. LV-SEM has the advantage of allowing observation at a high magnification using immunostained paraffin sections. In this study, LV-SEM was used to measure the length of infiltrating macrophages, which is difficult to measure with optical microscopy due to its low magnification. The length of macrophages in the DSF group was significantly shorter compared to the Vehicle group. In a previous study, we reported that macrophages infiltrating glomeruli were smaller in size in the group treated orally with DSF in a rat model of anti-GBM glomerulonephritis [21]. The results of this experiment also show a similar reduction in the size of infiltrating macrophages, which indirectly suggests that DSF may inhibit macrophage pseudopodia formation by inhibiting FROUNT.

In addition to FROUNT inhibition, DSF has been reported to target a variety of other molecules, including aldehyde dehydrogenase, nuclear protein localization protein 4 (NPL4), and gasdermin D. Inhibition of NPL4 is known to induce apoptosis by inhibiting nuclear factor-kB, and inhibition of gasdermin D blocks IL-1β release by preventing membrane pore formation [18,20]. The present results of reductions in the number of infiltrating macrophages and size of macrophages support the possibility that DSF inhibits macrophage infiltration via FROUNT; however, further studies on other target molecules are needed.

Since DSF eye drops suppressed macrophage infiltration, corneal scarring, and angiogenesis, DSF eye drops are likely beneficial for the long-term prognosis of vision after alkali burn. DSF eye drops are expected to be applied to improve the long-term prognosis of acute inflammatory diseases as well as chronic inflammatory diseases related to macrophages and granulomatous uveitis. Future studies are needed to compare the efficacy of DSF relative to steroid eye drops, and in combination with steroid eye drops. 

HE-staining images of the central cornea showed that the corneal epithelium was completely covered at 1 day after alkali burn. Usually, with alkali burns, in which corneal epithelial defects are severe and extensive, it is unlikely that the corneal epithelium is completely covered in less than 1 day. The complete coverage of the corneal epithelium after 1 day in this study may be due to the mild and controlled alkali burn procedure applied in our model for experimental stability. At 6 h, corneal epithelial defects remained in both groups, but the extent of the defects was smaller in the DSF group, suggesting that epithelial coverage may be more rapid in the DSF group. However, this is not a quantitative evaluation, and further research is needed.

In clinical practice, the acute phase of alkali burns may also be associated with severe melting and corneal perforation. The efficacy of DSF eye drops as a treatment for these conditions was not studied in this experiment and should be investigated in the future. In addition, limbal stem cell deficiency may occur in corneal alkali burns. It is necessary to extend the duration of the experiment to verify the effect of DSF on limbal stem cell deficiency.

## 4. Materials and Methods

### 4.1. Animals and Ethics Statement

Eight-week-old male Wistar rats from Sankyo Laboratory Service (Tokyo, Japan) were used for all experiments in this study. The breeding room was on a 12:12 h light/dark cycle. All rats were provided food ad libitum. All animal experiments were conducted in compliance with the Experimental Animal Ethics Review Committee of Nippon Medical School (approval number: 2020-100, 25 December 2020) and all procedures conformed to the Association for Research in Vision and Ophthalmic and Visual Research.

### 4.2. Experimental Procedures

Under isoflurane general anesthesia, a 3.2 mm diameter circular filter paper was soaked in 1 N NaOH and placed on the central cornea of each rat for 1 min to create corneal alkali burns. After alkali exposure, the ocular surface was washed with 40 mL of saline solution. Eye drops consisting of 0.5% DSF or vehicle were administered twice daily to the alkali burned cornea, beginning immediately after the saline wash. Each drop is one drop at a time, and the volume of one drop is about 50 µL. Eye drops were prepared based on a previous report of DSF eye drops [22]. For the vehicle eye drops, 5 g hydroxypropyl-β-cyclodextrin (HPβCD) and 0.1 g hydroxypropyl methylcellulose (HPMC) (both from FUJIFILM Wako Pure Chemical Corp, Osaka, Japan) were dissolved in 100 mL of saline; for 0.5% DSF eye drops, 0.5 g microbead-milled disulfiram (NOCBIN; Mitsubishi Tanabe Pharma Corp, Osaka, Japan), 5 g HPβCD, and 0.1 g HPMC were dissolved in 100 mL of saline. Benzalkonium chloride solution (Kozakai Pharmaceutical Corp, Tokyo, Japan) was added as a preservative to the prepared ophthalmic solutions to a concentration of 0.005%. The DSF concentration in the 0.5% DSF eye drop is 16.8 mmol/dm³. At each endpoint (6 h, and 1, 4, and 7 days after alkali burn), rats were sacrificed by exsanguination under isoflurane anesthesia. The removed eyes were evaluated for ocular pathology and molecular biology.

### 4.3. Histological and Immunohistochemical Analyses

The excised eyes were fixed in 10% buffered formalin and embedded in paraffin. For histopathological examination, EST staining was performed to detect infiltrating neutrophils. Primary antibodies used for immunohistochemical analysis were monoclonal mouse anti-rat ED-1 (BMA, Nagoya, Japan), polyclonal goat anti-FROUNT (Everest Biotech, Oxfordshire, UK), monoclonal rabbit anti-phosphorylated AKT (Cell Signaling Technology, Danvers, MA, USA), polyclonal goat anti-TNF-α (Santa Cruz Biotechnology, Dallas, TX, USA), polyclonal rabbit anti-TGF-β1 (Santa Cruz Biotechnology, Dallas, TX, USA), and polyclonal goat anti-IL-1β (R&D Systems, Minneapolis, MN, USA). Corneal scarring was evaluated with polyclonal goat anti-collagen type III (Southern Biotechnology, Birmingham, AL, USA). Angiogenesis was evaluated using monoclonal mouse anti-nestin (Merck Millipore, Darmstadt, Germany). Macrophage and neutrophil counts were observed at 400× magnification in three locations in the central cornea and two locations in the peripheral cornea. Cell counts were averaged separately for the central and peripheral corneas. Nestin-positive cells were averaged over two locations in the peripheral cornea at 400× magnification. Collagen type III-positive areas were observed at three locations in the central cornea at 400× magnification, and the ratio of collagen type III-positive area to corneal stromal area was calculated and averaged.

### 4.4. Real-Time RT-PCR

The mRNA expression of TNF-α, TGF-β1, IL-1β, and VEGF-A was examined using real-time RT-PCR. The corneas were excised from rat eyeballs and stabilized in RNAlater (Qiagen, GmbH, Hilden, Germany). Total RNA was extracted from the corneas using ISOGEN II (Nippon Gene, Tokyo, Japan). RNA concentration was measured using a NanoDrop ND1000 V3.2.1 spectrophotometer (Thermo Fisher Scientific, Waltham, MA, USA). cDNA was synthesized using the High Capacity cDNA Reverse Transcription kit (Thermo Fisher Scientific, Waltham, MA, USA). Real-time PCR was performed using the QuantStudioTM 3 Real-Time PCR System (Thermo Fisher Scientific, Waltham, MA, USA) and THUNDERBIRD SYBR qPCR Mix (TOYOBO, Osaka, Japan), and specific primers were used to amplify target genes (2 min at 50 °C, 10 min at 95 °C, and 45 cycles of denaturation at 95 °C for 15 s and annealing at 60 °C for 60 s). mRNA expression levels were normalized to that of β-actin. Primers used in this experiment are listed below (Table 1).

### 4.5. Low-Vacuum Scanning Electron Microscopy Imaging

Sections of paraffin-embedded rat corneal tissue were stained with ED-1 and visualized with DAB. A method using methenamine silver and gold chloride developed in our laboratory was used to enhance the signal of DAB staining [29]. Platinum blue was also used to enhance tissue contrast [26,27]. ED-1-positive macrophages infiltrating the peripheral cornea 1 day after alkali burn were observed using a low-vacuum scanning electron microscope (LV-SEM; Hitachi Tabletop Microscope TM3030Plus; Hitachi High- Technologies Corp., Tokyo, Japan) at 2000× to 8000×. The length of ED-1-positive cells was measured using LV-SEM images. Eleven eyes in each of the DSF and Vehicle groups were measured. The length of 392 cells was measured in each group.

### 4.6. Statistical Analysis

Statistical analysis was performed using an unpaired Student’s *t*-test. All results are expressed as mean ± standard error, and *p* < 0.05 was considered to indicate statistical significance. All analyses were calculated using GraphPad Prism software (Version 9.4.0, GraphPad Software, San Diego, CA, USA).

## 5. Conclusions

Disulfiram eye drops inhibited corneal infiltration of macrophages by obstructing pseudopodia formation, accompanied by abrogation of corneal scarring and neovascularization in alkali-burned corneas. These results suggest that disulfiram may be a new candidate therapeutic strategy for macrophage-associated corneal inflammatory diseases.

## Figures and Tables

**Figure 1 ijms-24-00735-f001:**
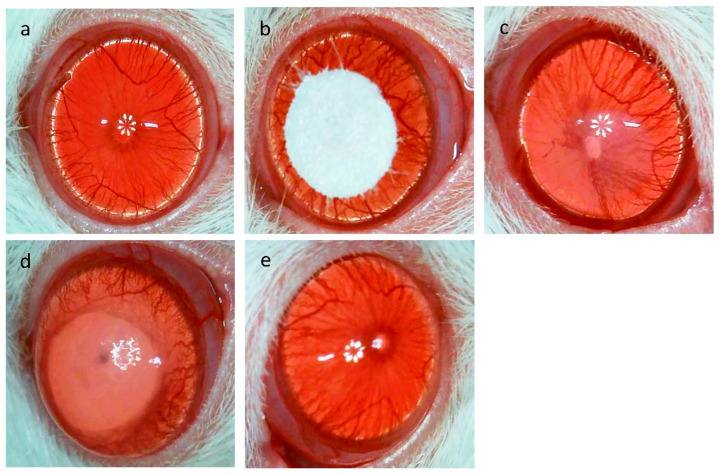
Images of the anterior eye of rats before and after alkali burn are shown. Before alkali burn (**a**). During alkali burn treatment (**b**). Immediately after alkali burn (**c**). At 7 days after alkali burn (Vehicle group) (**d**). At 7 days after alkali burn (disulfiram (DSF) group) (**e**). HE staining images of the central cornea, 6 h Vehicle group (**f**), 6 h DSF group (**g**), 1 day Vehicle group (**h**), 1 day DSF group (**i**). Bar, 500 μm. The black arrows indicate the terminus of the corneal epithelium.

**Figure 2 ijms-24-00735-f002:**
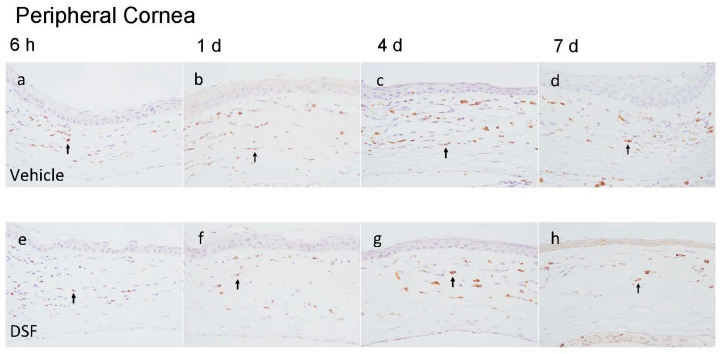
ED-1 staining was performed to evaluate macrophage infiltration in the rat cornea after alkali burn. Immunohistochemical images of the peripheral cornea at 6 h and 1, 4, and 7 days are shown (Vehicle group: (**a***–***d**), DSF group: (**e***–***h**)). Immunohistochemical images of the central cornea at 6 h and 1, 4, and 7 days are shown (Vehicle group: (**i***–***l**), DSF group: (**m***–***p**)). The number of ED-1-positive cells infiltrating at four endpoints in the Vehicle and DSF groups were compared. Peripheral corneas showed a statistically significant difference after 1 day (**q**). Central corneas showed a statistically significant difference after 4 days (**r**). Bar, 50 μm. The black arrows indicate ED-1-positive cells. Data are presented as mean ± standard error (*n* = 8 samples/group). “ns” means no significant difference. ** *p* < 0.01, * *p* < 0.05.

**Figure 3 ijms-24-00735-f003:**
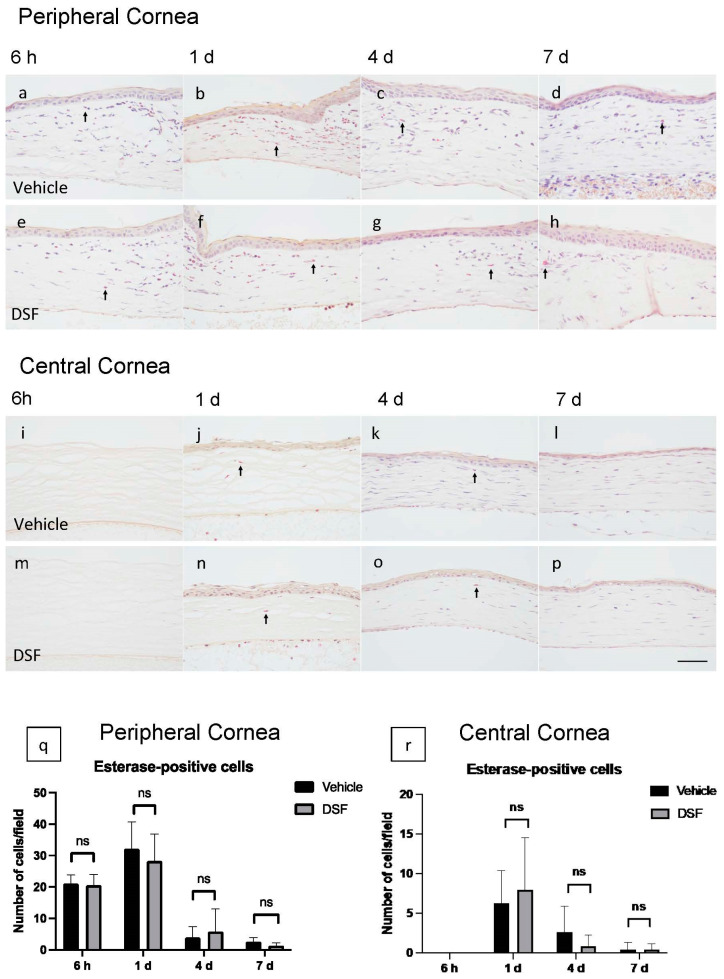
Esterase (EST) staining was performed to evaluate infiltrating neutrophils in the rat cornea after alkali burn. Immunohistochemical images of the peripheral cornea at 6 h and 1, 4, and 7 days are shown (Vehicle group: (**a***–***d**), DSF group: (**e***–***h**)). Immunohistochemical images of the central cornea at 6 h and 1, 4, and 7 days are shown (Vehicle group: (**i**–**l**), DSF group: (**m***–***p**)). The number of EST-positive cells infiltrating the four endpoints in the Vehicle and DSF groups were compared. In the peripheral cornea, there was no statistically significant difference in any of the endpoints (**q**). No statistically significant differences were also found in the central cornea (**r**). Bar, 50 μm. The black arrows indicate EST-positive cells. Data are presented as mean ± standard error (*n* = 8 samples/group). “ns” means no significant difference.

**Figure 4 ijms-24-00735-f004:**
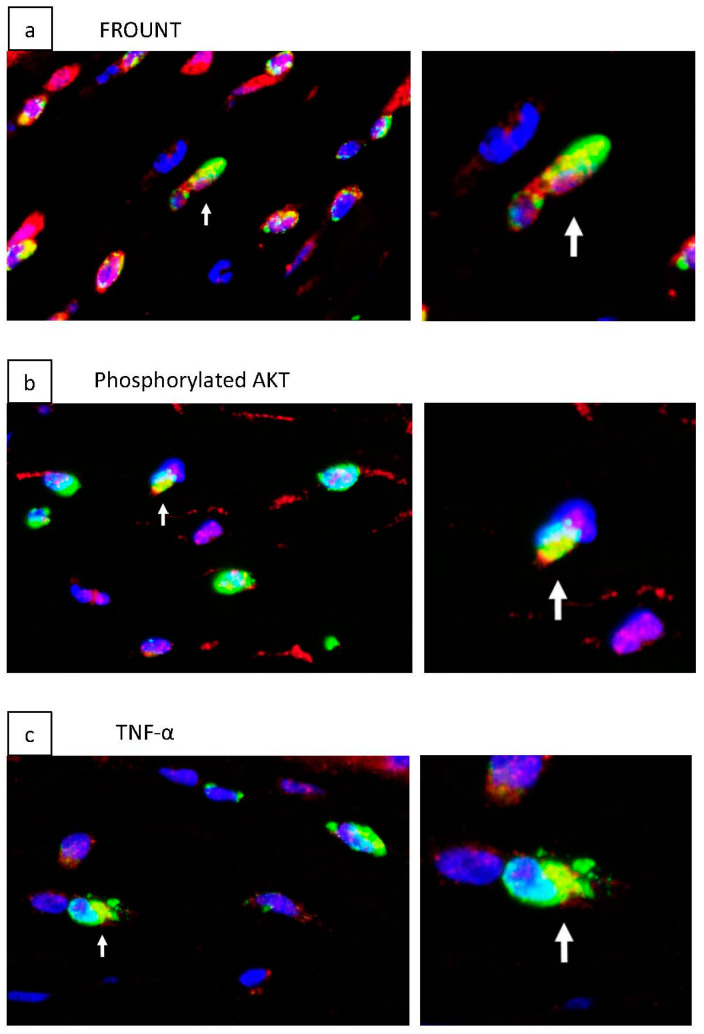
Fluorescence immunostaining was performed to determine the presence of FROUNT-related molecules and macrophage-associated cytokines on ED-1-positive cells infiltrated in a corneal alkali burn model. Fluorescence immunostaining was performed in the Vehicle group 1 day after alkali burn. Blue-stained cell nuclei with DAPI, green-stained ED-1, and red-stained target molecules, respectively, were evaluated. Areas that are positive for both ED-1 and the target molecule appear to be stained yellow. FROUNT was present on ED-1-positive cells (**a**). Phosphorylated AKT, a product of the FROUNT-induced PI3K pathway, was present on ED-1-positive cells (**b**). TNF-α, TGF-β1, IL-1β were also present on ED-1-positive cells (**c***–***e**). RT-PCR was performed on corneas 4 days after alkali burn, and macrophage-related cytokines of the DSF and Vehicle groups were compared. The results showed that TNF-α, TGF-β1, and IL-1β were significantly downregulated in the DSF group (**f***–***h**). Bar, 10 μm. White arrows indicate cells double positive for ED-1 and target molecules (yellow staining). Data are presented as mean ± standard error (*n* = 8 samples/group). * *p* < 0.05.

**Figure 5 ijms-24-00735-f005:**
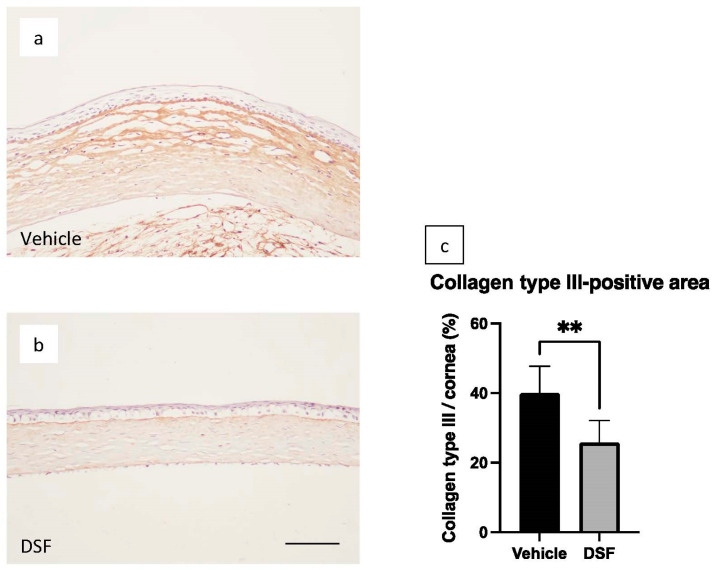
To evaluate central corneal scarring, immunohistochemical analysis of collagen type III in the Vehicle group (**a**) and DSF group (**b**) was performed in rat corneas 7 days after alkali burn. Comparison of the percentage of collagen type III expression area/corneal stroma area between the two groups showed a significantly lower percentage of collagen type III in the DSF group (**c**). Bar, 50 μm. Data are shown as mean ± standard error (*n* = 8 samples/group). ** *p* < 0.01.

**Figure 6 ijms-24-00735-f006:**
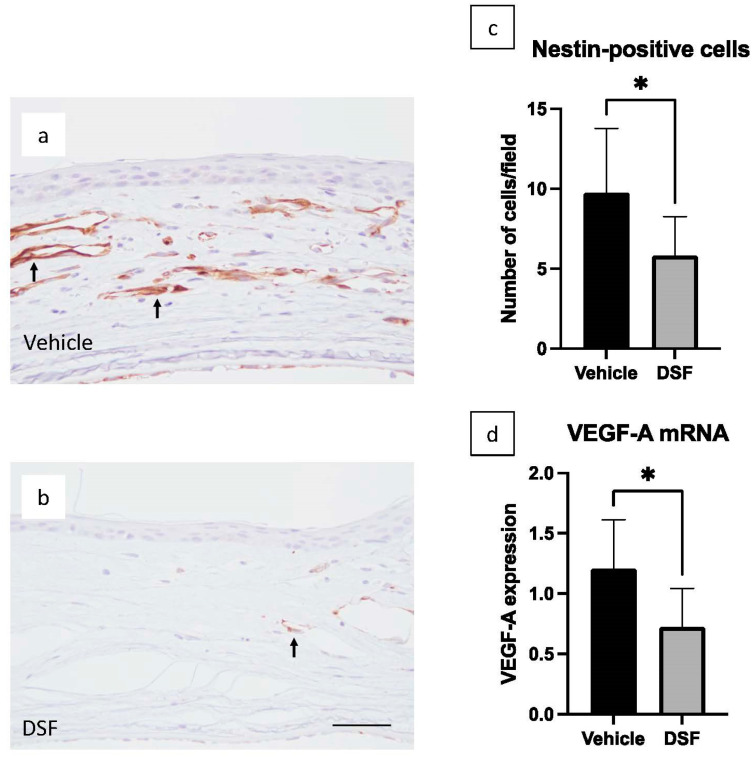
To evaluate neovascularization of the peripheral cornea, immunohistochemical analysis of nestin in the Vehicle group (**a**) and DSF group (**b**) was performed in rat corneas 4 days after alkali burn. Enumeration of nestin-positive vascular endothelial cells was performed and comparisons were made between the Vehicle and DSF groups (**c**). RT-PCR was performed in rat corneas 4 days after alkali burn to examine VEGF-A expression and comparisons were made between the Vehicle and DSF groups (**d**). Bar, 50 μm. The black arrows indicate nestin-positive cells. Data are shown as mean ± standard error (*n* = 8 samples/group). * *p* < 0.05.

**Figure 7 ijms-24-00735-f007:**
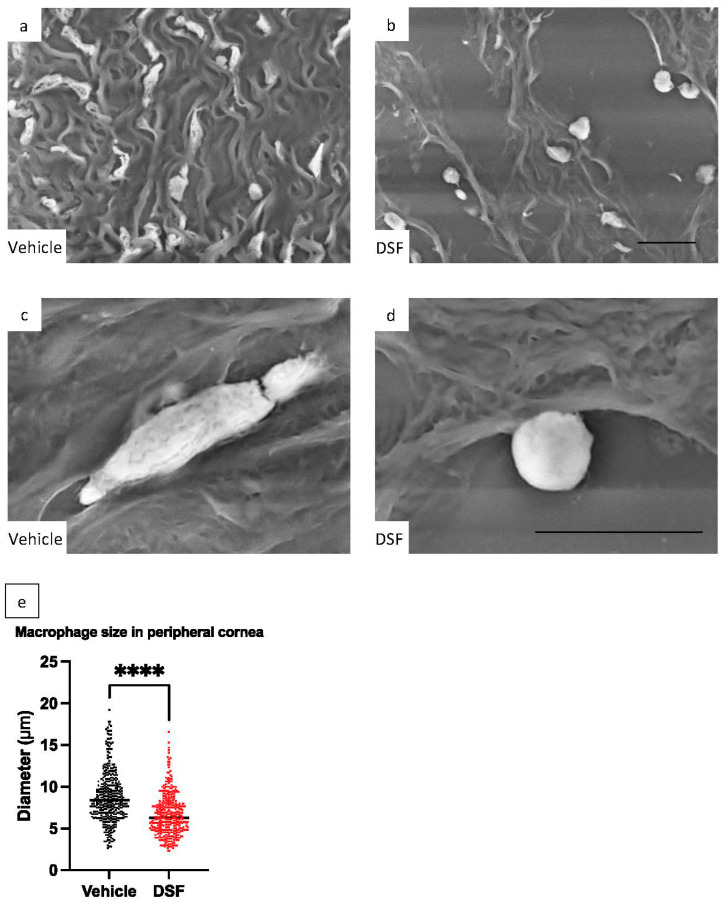
To evaluate pseudopodia formation of macrophages infiltrating the peripheral cornea in the Vehicle group (**a**,**c**) and DSF group (**b**,**d**), the length of ED-1-positive cells was evaluated in rat corneas using LV-SEM 1 d after alkali burn. Images (**a**,**b**) were taken at 2000*×*. Bar, 10 μm. Images (**c**,**d**) were taken at 8000*×*. Bar, 10 µm. The length of macrophages was measured. The length of 392 ED-1-positive cells from 11 eyes that had infiltrated the peripheral cornea in both groups was measured and compared (**e**). Data are shown as mean ± standard error (*n* = 392 samples/group). **** *p* < 0.0001.

**Table 1 ijms-24-00735-t001:** PCR primers used in this study.

Gene	Forward Primer Sequence (5′-3′)	Reverse Primer Sequence (5′-3′)
β-actin	GCAGGAGTACGATGAGTCCG	ACGCAGCTCAGTAACAGTCC
TNF-α	AAATGGGCTCCCTCTCATCAGTTC	TCTGCTTGGTGGTTTGCTACGAC
TGF-b1	TGGCCAGATCCTGTCCAAAC	GTTGTACAAAGCGAGCACCG
IL-1b	TACCTATGTCTTGCCCGTGGAG	ATCATCCCACGAGTCACAGAGG
VEGF-A	GCAGCGACAAGGCAGACTAT	GCAACCTCTCCAAACCGTTG

## Data Availability

The data presented in this study are available on reasonable request from the corresponding author.

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
