# Peer review of "Disulfiram Ophthalmic Solution Inhibited Macrophage Infiltration by Suppressing Macrophage Pseudopodia Formation in a Rat Corneal Alkali Burn Model"

_ijms, 2023, doi:10.3390/ijms24010735_

Round 1
Reviewer 1 Report
Dear editor, first I would like to commend the authors for the effort in delivering this manuscript on the challenging topic of corneal complications secondary to chemical burns. In special, for introducing an additional drug (DFS) to the therapeutic armamentarium for this severe condition. In order to improve the paper readability and impact I have some questions and comments to the authors.
Results.
The grammar here should be revised. Most sentences in section 2.1 are in the present tense, while in other sections they are in the past tense. I suggest the authors use the past tense for the sake of consistency.
METHODOLOGY
The methodology used by authors was based on the article by Choi et al. in which only corneal opacity and neovascularization are evaluated. However persistent corneal epithelial defects are a hallmark of chemical burn clinical presentation, hence objective measurement of the ulcer should also have been assessed and compared with controls. Proper methodology for conducting this analysis can be found in:
"Tong, L.; Png, E.; Aihua, H.; Yong, S.S.; Yeo, H.L.; Riau, A.; Mendoz, E.; Chaurasia, S.S.; Lim, C.T.; Yiu, T.W.; et al. Molecular mechanism of transglutaminase-2 in corneal epithelial migration and adhesion. Biochimica et biophysica acta 2013, 1833, 1304-1315, doi:10.1016/j.bbamcr.2013.02.030."
"Bandeira, F.; Goh, T.W.; Setiawan, M.; Yam, G.H.; Mehta, J.S. Cellular therapy of corneal epithelial defect by adipose mesenchymal stem cell-derived epithelial progenitors. Stem cell research & therapy 2020, 11, 14, doi:10.1186/s13287-019-1533-1."
Another important feature of chemical burns is stromal melting with loss of corneal tissue, which the authors also lacked to show. Corneal melting is strongly associated with up/downregulation of inflammatory biomarkers present in the ocular surface such as MMP and interleukins (IL-1, IL-6, IL-11) and tissue edema and subsequent loss can also be clinically measured with OCT as described in:
Bandeira, F.; Goh, T.W.; Setiawan, M.; Yam, G.H.; Mehta, J.S. Cellular therapy of corneal epithelial defect by adipose mesenchymal stem cell-derived epithelial progenitors. Stem cell research & therapy 2020, 11, 14, doi:10.1186/s13287-019-1533-1.
Yam, G.H.; Fuest, M.; Yusoff, N.; Goh, T.W.; Bandeira, F.; Setiawan, M.; Seah, X.Y.; Lwin, N.C.; Stanzel, T.P.; Ong, H.S.; et al. Safety and Feasibility of Intrastromal Injection of Cultivated Human Corneal Stromal Keratocytes as Cell-Based Therapy for Corneal Opacities. Invest Ophthalmol Vis Sci 2018, 59, 3340-3354, doi:10.1167/iovs.17-23575.
Regarding the histological studies: Immunofluorescence staining is more effective and accurate in the detection of the target molecules within the sectioned cornea. Why did the authors opt to use immunohistochemical analysis instead? Is there any methodological explanation for this decision?
INHIBITION OF SCARRING
Corneal stromal scars are usually the result of stromal keratocyte activation into fibroblasts and subsequent transformation into myofibroblasts. Why the authors have chosen to analyze expression of collagen III (one of the final byproducts of corneal scarring) as a main outcome in their study? Are there any methodological reasons why they did not investigate the presence of vimentin and alfa-SMA?
DISCUSSION
The discussion is mostly focused on the mollecular basis of corneal wound healing after the chemical burn. However there is little, if any, translation to the clinical importance of the actual findings. The natural history of chemical burn and consequently its treatment can be divided in two phases: first the hyperacute, in which there is a great risk of severe melting and corneal perforation, with subsequent need of therapeutic keratoplasty along with all the risks of an open-sky emergency procedure. Since the timeframe of the study is exactly within the acute/hyperacute phase, the authors should point out if DSF can be useful in preventing this complication and explain how.
Another key point to be discussed is the neovascularization and limbal stem cell deficiency, which are commonly associated. These complications may not necessarily occur within the first days of the lesion, and may arise within weeks.
The authors have demonstrated an important antiangiogenic effect of DSF which is clearly noted in the slides comparison of nestin + cells and RT-PCR of VEGF-A (Fig. 6). However, deep neovascularization is not usually noted within the first days of the wound in humans, this finding is only present at later phases. These results can be a feature inherent to the wound model (thinner cornea), hence the antiangiogenic effect may not translate clinically to humans. Can the authors comment on these points?
The timeframe of the study does not allow readers to properly assess if the anti-inflammatory effect of DSF can be useful in preventing LSCD, a longer timeframe of DSF use and follow up specifically focused on LSCD markers would address this issue - this drawback should be mentioned in the study.
Reviewer 2 Report
Authors have investigated disulfiram eye drops to treat corneal alkal burn injury. It reduced macrophage infiltration and be potential treatment option for corneal inflammation.
Abstract
- open all abbreviations when mentioned first time e.g. FROUNT, CCR5.
- give short introduction what means ”alkali burn model”
Results
- Figure 4, is it also possible to get enlarged image of site of arrow with original image to see staining results better.
- please explain more Figure 4 staining there was in text mentioned that red is stained molecule and then in figure legend that yellow. make more clear.
- Explain little bit more figure 5 or indicate by arrow what differences there is seen in a and b.
Materials and methods
- Describe eye drop more detailed. One drop? What is estimated volume of drop and concentration of DSF?
Reviewer 3 Report
The present study is innovative and demonstrate the beneficial effects of Disulfiram in preventing corneal blindness by decreasing macrophages infiltration and subsequently, decreasing scar tissue and angiogenesis. The authors addressed all the contributing mechanisms of corneal blindness.
